# Changes of Volatile Organic Compounds of Different Flesh Texture Pears during Shelf Life Based on Headspace Solid-Phase Microextraction with Gas Chromatography–Mass Spectrometry

**DOI:** 10.3390/foods12234224

**Published:** 2023-11-23

**Authors:** Yuqing Xu, Guanwei Gao, Luming Tian, Yufen Cao, Xingguang Dong, Hongliang Huo, Dan Qi, Ying Zhang, Jiayu Xu, Chao Liu

**Affiliations:** 1Institute of Pomology, Chinese Academy of Agricultural Sciences, Xingcheng 125100, China; xuyuqing0325@163.com (Y.X.);; 2Key Laboratory of Germplasm Resources Utilization of Horticultural Crops, Ministry of Agriculture and Rural Affairs, Xingcheng 125100, China

**Keywords:** pear, volatile organic compounds, flesh texture, shelf life, HS-SPME, GC-MS

## Abstract

Aroma is an important sensory factor in evaluating the quality of pear fruits. This study used headspace solid-phase microextraction (HS-SPME) combined with gas chromatography–mass spectrometry (GC-MS) to analyze the volatile organic compounds (VOCs) of three crispy pears and five soft pears during shelf life, and the changes in soluble solids content (SSC) were analyzed. The results showed that the SSC of the soft pears such as Nanguoli, Jingbaili and Louis was always higher than that of the crispy pears throughout shelf life. A total of 160 VOCs were detected in the eight pear varieties. Orthogonal partial least squares discriminant analysis (OPLS-DA) and hierarchical cluster analysis (HCA) combined with predictor variable importance projection (VIP) showed that the eight pear varieties could be obviously classified into six groups according to the differences in their VOCs, and 31 differential VOCs were screened out, which could be used to differentiate between pears with different flesh textures. The results of clustering heat map analysis showed that, with the extension of shelf life, the content of each different VOC did not change much in crispy pears, whereas the difference in soft pears was larger. This study confirmed the potential of determining the optimal shelf life of different pear varieties about aroma evaluation and studying the mechanism of differences in VOCs in the future.

## 1. Introduction

Pear (*Pyrus*) is one of the most important temperate fruit tree species in the world, with a global cultivation area of about 1.4 million hm^2^ and a total annual production of about 25.66 million tons [1], which has outstanding economic benefits. Aroma is an important indicator reflecting the flavor, ripeness and quality of pear fruits. The pleasant aroma of pear fruits is also a crucial factor in attracting consumers and enhancing market competitiveness [2]. Pear varieties are divided into European and Asian pears due to their geographical distribution [3]. Most European pears have a soft and juicy flesh texture and strong aroma, and main cultivars belong to *P. communis* [2,4]. However, most *P. ussuriensis* cultivars share similarities with European pears in softer fruits and develop very strong aroma during their ripening period [5]. The main cultivars of Asian pears belong to *P. bretschneideri*, *P. pyrifolia*, *P. ussuriensis* and *P. sinkiangensis*, and most crispy Asian pears have a crisp flesh with more juice but light aroma [4,6]. In recent years, there have been many research reports on pear fruits aroma and VOCs. The VOCs of pears have been studied, and more than 300 kinds of VOCs have been detected in pear fruits [7]. Aroma consists of a mixture of VOCs, mainly including esters, aldehydes, alcohols, ketones and terpenoids [8,9,10]. Various combinations and contents of VOCs in fruits contribute to significant differences in aroma [11].

Although the type and content of VOCs can be influenced by some factors, such as cultivation [12,13,14], harvest maturity [15,16,17] and storage conditions [18,19,20,21], variety itself is the fundamental factor affecting aroma formation. Furthermore, the formation of aroma substances is a dynamic process during fruit growth and storage. There are significant differences in the composition and concentration of VOCs in different varieties: esters are the most important volatile compounds in the fruit of *P. communis* [9]; the volatile compounds of *P. ussuriensis* are mainly esters, hydrocarbons and aldehydes [22]; *P. pyrifolia* mainly has esters, alcohols and aldehydes [10]; the volatile compounds of *P. bretschneideri* are mainly esters and alcohols [23]; and the main volatile compounds of Korla pear (*P. sinkiangensis*) are hexanal, (E)-2-hexenal, 1-hexanol, (E)-2-hexen-1-ol, (Z)-3-hexen-1-ol and hexyl acetate [17]. Moreover, it was found that the VOCs of wild Kainth (*P. pashia*) pears in India are mainly esters and alkanes [24]. In addition, the quality of pear fruits is also influenced by the length of the storage period. Researchers found that the content of esters in the Dr. Jules Guyot pear gradually increased and the content of alcohols and aldehydes gradually decreased as the storage time increased [10].

In recent years, HS-SPME combined with GC-MS has been used to accurately and efficiently analyze and identify volatiles in fruits [25]. It has been widely used for the determination of VOCs in apples [26,27], pears [28,29], grapes [30,31], peaches [32], citrus [33] and other fruits. Currently, important progress has been made in research on pear flavor quality, but relatively few studies have been conducted on the changes in the VOCs of pear varieties with different flesh textures during different post-harvest shelf life. Storage is very important for pear fruits, not only to extend the freshness period of pear fruits and ensure their quality and taste, but also to increase the economic value of the fruits. In addition, the length of shelf life affects the type and content of aroma substances in the fruits, and changes in aroma substances are important in determining the shelf life of fruits. In this experiment, HS-SPME with GC-MS was used to study the differences in VOCs among eight pear varieties during their shelf life at room temperature. The aims of this study were to provide a certain theoretical basis for the evaluation and analysis of the aroma quality of pear fruits during their shelf life, as well as to provide a reference to determine the best aroma sensory quality during shelf life, according to the aroma characteristics of the pear fruits, and a mechanism of study of the differences of VOCs in different pears.

## 2. Materials and Methods

### 2.1. Subsection Chemicals and Reagents

The internal standard cyclohexanone (>99%) was provided by Tianjin Chemical Reagent Ltd. (Tianjin, China). The standard solution was prepared with 10% methanol, which was obtained from Fisher Scientific (Pittsburgh, PA, USA) (HPLC grade). Pure water was purchased from Wahaha Foods Co., Ltd. (Hangzhou, China). Sodium chloride (NaCl) was supplied by Agela Technologies (Newark, DE, USA).

### 2.2. Fruit Materials

On 22 September 2022, eight pear varieties with different flesh texture characteristics and similar maturity were selected from the “National Pear and Apple Germplasm Resource Repository (Xingcheng)”. They were the soft-flesh pear varieties of Nanguoli (*P. ussuriensis*), Jingbaili (*P. ussuriensis*), Ruanerli (*P. ussuriensis*), Cure (*P. communis*) and Louis (*P. communis*), and the crisp pear varieties of Korla pear (*P. sinkiangensis*), Yali (*P. bretschneideri*) and Hanhongli [Nanguoli × (Yali × Jinli)] as test materials. Fruits of uniform size and maturity without pests and mechanical damage were picked and placed in the laboratory. The fruits were stored at room temperature, and samples were taken to determine the composition and concentration of VOCs at 0 d, 3 d, 7 d, 14 d and 21 d of shelf life at room temperature. Each sample was analyzed and determined in triplicate.

### 2.3. Sample Preparation

For sample preparation, after removing the peel and core, the flesh was sliced with a stainless-steel knife, and the flesh was mixed with NaCl (1:1, m/m) and homogenized in a homogenizer. Subsequently, the homogenized mixture (10 g) was placed into a 20 mL SPME vial, after which 100 μL of cyclohexanone standard solution (0.2 mg/mL) was added to the vial, which was fixed to 10 mL with purified water, and the vial was immediately sealed with a screw cap with a polytetrafluoroethylene/silicone septum. All samples were stored at −20 °C until analysis.

### 2.4. Determination of Soluble Solids Content

A total of 10 fruits were taken from each variety, the flesh was sliced and mixed after removing the peel and core, and the appropriate amount of liquid was directly squeezed and dripped onto a PAL-1 digital refractometer (ATAGO, Tokyo, Japan) to determine the SSC (%) of the pears. Ten replicates were determined and the mean value was recorded.

### 2.5. HS-SPME Conditions

VOCs were extracted using a divinylbenzene/carboxene/polydimethylsiloxane (DVB/CAR/PDMS) composite extraction head, which was purchased from Sigma-Aldrich (Supelco, Bellefonte, PA, USA). The VOCs were extracted from the samples using an AOC 6000 autosampler (Shimadazu, Tokyo, Japan). The samples were incubated at 80 °C for 15 min, followed by headspace extraction for 15 min, and the extraction head was fed into the GC-MS injection port.

### 2.6. GC-MS Analysis

The samples were analyzed using a Shimadazu 2010plus GC equipped with a QP2010 mass spectrometry system (Shimadazu, Tokyo, Japan). The VOCs on the extraction head were desorbed at the injector port for 1 min at 200 °C using the splitless mode.

GC conditions: the HP-INNOWAX capillary column (60 m × 0.25 mm × 0.25 μm, Agilent Technologies, Santa Clara, CA, USA) was used for the separation. The initial column temperature was 50 °C and held for 1 min, then increased to 180 °C at 2 °C /min and held for 1 min, and finally increased to 230 °C at 10 °C /min and held for 10 min. Helium was used as the carrier gas at a constant flow rate of 1.0 mL/min (purity not less than 99.999%).

MS conditions: the ion source was an electron ionization (EI) source with an ion source energy of 70 eV, and the temperature of the ion source in the detector was 230 °C. The mass spectrum was in full scan mode with a scanning range of *m*/*z* 50–500.

The total ion flow chromatogram was obtained based on GC-MS analysis, and the metabolites of the samples were characterized by mass spectrometry based on the NIST 17s database. Only volatiles with a match of 90% or more were retained, and the corresponding CAS registry numbers and retention times (RT) were queried. All data were obtained from experimental results in triplicate.

### 2.7. Statistical Analysis

Based on the peak area normalization method, the VOC content was obtained by comparing the peak area with that of the internal standard (cyclohexanone), and the content of VOCs (mg/kg FW) = (peak area of VOCs/peak area of the internal standard) × content of the internal standard. Statistical analyses and bar graphs were performed for the number and content of volatile components using Excel version 2016 (Microsoft, Redmond, WA, USA). Origin 2021 software (OriginLab Corporation, Northampton, MA, USA) was used to plot correlation heat maps. Heat maps were plotted by TBtools version 1.09876 (Guangzhou, China). SIMCA version 14.1 (Umetrics, Umea, Sweden) was used to perform the principal component analysis (PCA), HCA, OPLS-DA and VIP. And one-way analysis of variance (ANOVA) was performed with SPSS version 26.0 (International Business Machines Corporation, Armonk, NY, USA); *p* < 0.05 was considered significant.

## 3. Results and Discussion

### 3.1. Changes in SSC of Eight Pear Varieties during Shelf Life

Among crisp-flesh pears, the Korla pear flesh SSC was lower than the Hanhongli only in the shelf life of 14 d; in the rest of shelf life, the SSC values were Korla pear > Hanhongli > Yali, as seen in Table 1. During shelf life, the SSC values of soft-flesh pears in the Nanguoli, Jingbaili and Louis varieties were higher than the crispy Korla pear, Yali and Hanhongli. The SSC of Nanguoli was always in the higher level, and reached its maximum value in the shelf life of 7 d. The SSC of Ruanerli was only lower than Korla pear in the shelf life of 21 d; in the rest of shelf life, its SSC values were higher than the crisp-flesh pears. The SSC of Cure was always lower than Korla pear and Hanhongli during shelf life, but higher than Yali.

### 3.2. Analysis of Differences in the Composition of Total Volatile Compounds

It was previously found that esters were the VOCs with the highest number of compounds in pear fruits, with alcohols second only to esters [6,9]. Similar results were obtained in this study. A total of 160 VOCs were detected from Korla pear, Yali, Hanhongli, Nanguoli, Jingbaili, Ruanerli, Cure and Louis during the whole shelf life, and the detected VOCs were classified into seven major categories according to the difference of the main functional groups of the compounds, including 61 esters, 39 alcohols, 18 aldehydes, 12 ketones, 7 acids, 13 hydrocarbons, and 10 other compounds, as seen in Figure 1A. Nanguoli and Jingbaili had the most types of VOCs, namely, 81 kinds, whereas Cure had the fewest types of VOCs, namely, 41 kinds. Yali, Hanhongli, Nanguoli, Jingbaili and Ruanerli had the most types of esters, Cure and Louis had the most types of alcohol, Korla pear had the most alcohol and aldehydes, and these eight pear varieties had fewer ketones, acids, hydrocarbons and other compounds. In addition, only one kind of ester was detected in Cure, and no hydrocarbon was detected in Hanhongli.

There were significant differences in the content of VOCs among pear varieties, as seen in Figure 1B. The VOC content of the eight pear varieties was ranked as Nanguoli (87.43 mg/kg) > Ruanerli (83.50 mg/kg) > Jingbaili (51.10 mg/kg) > Korla pear (30.62 mg/kg) > Yali (27.79 mg/kg) > Hanhongli (27.21 mg/kg) > Louis (23.77 mg/kg) > Cure (21.69 mg/kg). The relative content of VOCs in the eight pear varieties varied greatly among the varieties. The most VOCs in Ruanerli and Nanguoli were esters, and their relative content was 50.71% and 62.84%, much higher than the content of esters in the rest of the pear varieties. Korla pear, Yali, Jingbaili, Cure and Louis had high levels of aldehydes, with respective relative contents of 71.74%, 38.65%, 38.52%, 47.86% and 44.08%. The highest content of esters (35.07%) was found in Hanhongli among these crispy pears. In conclusion, the types and contents of VOCs in the eight pear varieties were relatively higher for esters, alcohols and aldehydes, compared to ketones, acids, hydrocarbons and other compounds, which were at a lower level. Other researchers found that esters, alcohols and aldehydes were the major VOCs in the fruits of 202 mature pear varieties [29]. Esters, alcohols and aldehydes were equally dominant in 33 *P. ussuriensis* cultivars [22].

### 3.3. Correlation Analysis between SSC and Total Content of Each Volatile Compound

The SSC, the ester content and the hydrocarbon content of pear varieties showed highly significant positive correlation with each other, as seen in Figure 2. This indicated that the higher the SSC of pears, the higher the content of esters and hydrocarbons contained in their flesh. Ester content was highly significantly positively correlated with alcohol content. Alcohol content was significantly positively correlated with aldehyde content, and aldehyde content was highly significantly positively correlated with acid content. The content of other compounds showed a significant positive correlation with the content of esters and aldehydes and a highly significant positive correlation with the content of alcohols.

### 3.4. Changes of VOCs in the Flesh of Eight Pear Varieties at Different Shelf Life

A total of 18 shared VOCs were simultaneously detected in the flesh of the eight pear varieties, as seen in Figure 3. These included two alcohols: 1-Hexanol and 2-Ethylhexanol; one ester: 2,2,4-Trimethyl-1,3-pentanediol diisobutyrate; seven aldehydes: Hexanal, (E)-2-Hexanal, Octanal, (E)-2 Heptenal, Nonanal, (E)-2-Octenal, (E)-2-Nonenal; three ketones: 1-Hepten-3-one, 6-Methyl-5-heptene-2-one, (E)-1-(2,6,6-Trimethyl-1,3-cyclohexadien-1-yl)-2-buten-1-one; three acids: Hexanoic acid, Octanoic acid and Nonanoic acid; and two other compounds: trans-α,α,5-Trimethyl-5-ethenyltetrahydro-2-furanmethanol and 3,5-bis(1,1-Dimethylethyl)phenol.

Aroma is a complex mixture of multiple VOCs [10], and the flavor of pears is closely related to the content and relative proportion of VOCs [34]. The types and contents of VOCs in different pear varieties changed in the course of shelf life (Figure 3). Throughout shelf life, the VOCs in Korla pear were mainly alcohols and aldehydes, and the high concentration of aldehydes and alcohols and the low concentration of esters contributed to a unique “fresh green” flavor [17]. The esters, alcohols and aldehydes were dominant in Yali and Hanhongli. This was similar to the results obtained in Laiyang pear (*P. bretschneideri*) [35]. With the prolongation of shelf life, the content of alcohols in Yali was always higher than that of Korla pear and Hanhongli, while the content of alcohols in Korla pear was lower than that of Hanhongli at shelf life of 0–7 d and higher than that of Hanhongli at shelf life of 14–21 d, as seen in Table 2. The aldehyde content of Korla pear flesh during shelf life was always higher than Yali and Hanhongli, while aldehyde content in Hanhongli was only higher than that of Yali during the shelf life of 14 d, and was lower than that of Yali in the rest of shelf life. Ester content in Hanhongli was lower than that of Yali only in the shelf life of 0 d and 21 d, and higher than that of Yali in the rest of shelf life. The content of the rest of the compounds in the three crisp-flesh pears was at a low level throughout shelf life.

Among the soft-flesh pears, the VOCs in Nanguoli, Jingbaili and Ruanerli were mainly esters, alcohols and aldehydes during the whole shelf life, as seen in Figure 3. Hydrocarbons were also the main VOCs of Nanguoli and Jingbaili during the shelf life of 14 and 21 d. Other researchers observed similar results that esters, alcohols, aldehydes and alkenes were the main aroma constituents in seven *P. ussuriensis* cultivars, including Anli, Jingbaili, Nanguoli and so on [34]. Esters and aldehydes were the key VOCs shared by 33 Chinese *P. ussuriensis* cultivars [22]. The VOCs in Cure and Louis were mainly alcohols, aldehydes and ketones throughout shelf life. Moreover, esters were also the major VOCs in Louis flesh at 21 d of shelf life. In a previous study, it was noted that the main VOCs in European pear varieties were esters, alcohols and alkanes [9]. With the extension of shelf life, among the five soft-flesh pears, the alcohol content in Ruanerli was always at a higher level during the shelf life of 0–14 d, and was lower than that of Nanguoli and Jingbali during the shelf life of 21 d in Table 2. During the whole shelf life, the alcohol content in Ruanerli, Nanguoli and Jingbaili was consistently higher than that of Louis and Cure. The aldehyde content in Jingbaili and Ruanerli was at a high level throughout the shelf life. The ester content in Ruanerli was always at a higher level during the shelf life of 0–7 d, but lower than Nanguoli in the shelf life of 14–21 d. The ketone content in Cure was higher than that of Louis during the shelf life of 0–7 d, but lower than Louis during the shelf life of 14–21 d.

### 3.5. Analysis of Key VOCs in Eight Pear Varieties Based on OPLS-DA and HCA

OPLS-DA is a multivariate calibration method [36] that maximizes intergroup differentiation and facilitates the search for differential metabolites [37]. In order to investigate the significantly different VOCs of different pear varieties at different shelf life, 160 VOCs of the eight pear varieties were used as dependent variables, and based on the content of aroma components of the eight pear varieties at different shelf life as independent variables, the eight pear varieties could be clearly classified into six groups by OPLS-DA and HCA (Figure 4A,B). Group 1 consisted of Nanguoli, group 2 consisted of Ruanerli, group 3 consisted of Louis and Cure, group 4 consisted of Korla pear, group 5 consisted of Jingbaili, and group 6 consisted of Yali and Hanhongli, to achieve the effective differentiation between the crisp-flesh pear and soft-flesh pear samples, indicating significant differences in the composition of VOCs between crispy and soft-flesh pears. After 200 replacement tests, the replacement test model R2 was 0.55, Q2 was −0.842, and the intersection point of the Q2 regression line and the vertical axis was less than 0 (Figure 4C), indicating that the discriminant model had a good explanatory ability and did not overfit, and the model validation was effective [38].

Variable importance in projection (VIP) in OPLS-DA represents the effect of differences between the corresponding metabolites on the classification and identification of each group of samples in the model, and it is generally accepted that there are significant differences in metabolites with VIP > 1 and *p* < 0.05 [39]. According to the OPLS-DA model, 32 differential aroma components with VIP >1 were screened (Figure 4D), and combined with *p* < 0.05 (Appendix A), a total of 31 differential VOCs were screened. Among them, there were 13 esters, 7 alcohols, 5 aldehydes, 3 ketones, 2 hydrocarbons and 1 other compound, which were Ethyl caproate, Hexanal, (E,E)-α-Farnesene, trans-α,α,5-Trimethyl-5-ethenyltetrahydro-2-furanmethanol, (E)-2-Hexanal, Ethyl butanoate, Ethyl (E,Z)-2,4-decadienoate, (E)-1-(2,6,6-Trimethyl-1,3-cyclohexadien-1-yl)-2-buten-1-one, (E)-2-Heptenal, 1-Hexanol, Butyl acetate, (E)-3-Hexenol acetate, Methyl caproate, (Z,E)-methyl 2,4-decadienoate, trans-3-Hexenol, α-Terpineol, (E)-2-Octenal, DL-2-Methylbutanol, 6-Methyl-5-heptene-2-one, Methyl butanoate, 1-Hepten-3-one, 1-Octen-3-ol, Ethyl 3-hydroxyhexanoate, Ethyl 2-methylbutanoate, Ethyl (E)-2-octenoate, Linalool, (Z,E)-α-Farnesene, (E)-2-Decenal, Hexyl acetate, Phenethyl acetate and (E)-2-Hexen-1-ol.

### 3.6. Discriminant Analysis of Flesh Texture of Pear Varieties Based on Differential VOCs

PCA is a common multivariate statistical method that can reflect the overall differences and variability between groups of samples from the data as a whole by reducing the dimensionality of the data set and simplifying the data structure [30,40]. PCA has been widely used to study the volatile components of pears [10,41]. In order to further analyze the differences of VOCs in pear varieties at different shelf life, based on the screening of 31 VOCs with differences, PCA was used to discriminate the aroma data of the eight pear varieties at different shelf life. The variance contribution rates of PC1 and PC2 were 44.9% and 21.8%, respectively, and the cumulative variance contribution rate reached 66.7%, which was considered to be a good reliability for selecting PC1 and PC2 analysis samples, as seen in Figure 5. Among them, Louis and Cure were negatively correlated with PC2, there was an overlap between the two, and the distinction was not significant. There was an overlap between Yali and Hanhongli, and the difference in volatile substances between the two was relatively small. Korla pear was negatively correlated with PC1, and Ruanerli was positively correlated with PC2.

The results showed that there were some differences in the VOCs of the pear varieties at different shelf life. Among the crisp-flesh pears, the differences of VOCs in Yali and Hanhongli were relatively small, while there was a separation between Korla pear and the other two crisp-flesh pears, and there were some differences of VOCs in the flesh. Among the soft-flesh pears, Louis and Cure were closer to each other, and the differences in VOCs were relatively small, while Nanguoli, Jingbaili and Ruanerli were all separated from the rest of the soft-flesh pears in an obvious way. From the results of the principal component analysis, the effect of differentiation between pear samples was poor and could not clearly reflect the differences between samples, so the differences in VOCs in pear samples were further explored using clustering heat map analysis.

### 3.7. Analysis of Differential VOCs in the Flesh of Eight Pear Varieties

The content of differential aroma components had an important contribution to the formation of the aroma characteristics of each variety, and the clustering heat map could clearly distinguish the higher content of differential VOCs in each texture, making the comparison of the content of differential VOCs in different pear varieties more obvious (Figure 6).

Esters. In terms of type and content, esters with a fruity flavor are the main VOCs in pears [10]. The ester content in the soft-flesh pears was larger and more variable than in the crisp-flesh pears. Jingbaili had a high concentration of Methyl butanoate and Methyl caproate. Both compounds exhibited an increasing and then decreasing trend with the prolongation of shelf life, with the highest content detected on the 14 d of shelf-life. Throughout the shelf life, Ruanerli and Nanguoli had high levels of Hexyl acetate, Butyl acetate, Ethyl caproate, Ethyl butanoate and Methyl caproate. Except for Butyl acetate, the other four compounds showed an increasing and then decreasing trend in Ruanerli, and an increasing and then decreasing trend in Nanguoli. Nanguoli had high content of Ethyl 3-hydroxyhexanoate, Ethyl 2-methylbutanoate and Ethyl (E)-2-octenoate, all of which were detected in the highest content at the 21 d of shelf life. (Z,E)-methyl-2,4-decadienoate and Ethyl (E,Z)-2,4-decadienoate were two esters with a pear-like odor [6], which were considered to be characteristic compounds of *P. communis* [9]. High levels of (Z,E)-methyl-2,4-decadienoate, Ethyl (E,Z)-2,4-decadienoate were also detected in Louis at the 21 d of shelf life. Nanguoli at the 14 and 21 d of shelf life had high concentrations of Ethyl (E,Z)-2,4-decadienoate.

Alcohols and aldehydes. C6 compounds (C6 alcohols and C6 aldehydes) are the representatives of the green aroma type of compounds, which contribute the green aroma odor to the fruits [42]. The volatiles in Yali, Hanhongli, Korla Pear, Jingbaili, Ruanerli and Nanguoli had higher levels of 1-Hexanol, Hexanal and (E)-2-Hexanal, and lower levels in Louis and Cure, as seen in Figure 6.

With the prolongation of shelf life, the 1-Hexanol content gradually decreased in Yali, had little change in Jingbaili, and showed the trend of increasing and then decreasing in Hanhongli, Korla pear, Ruanerli and Nanguoli. The Hexanal content showed the trend of increasing and then decreasing in Korla pear, and reached the maximum value in the 14 d of shelf life. Its content in Yali, Hanhongli, Ruanerli and Nanguoli was in a fluctuating state, while Jingbaili showed a trend of first decreasing and then increasing. Except for Yali, which reached the maximum value at the 7 d of shelf life, its values in the other four pear varieties were highest at 0 d of shelf life. (E)-2-Hexanal content in Yali, Hanhongli, Korla pear, Jingbaili and Nanguoli all showed the trend of first increasing and then decreasing with the prolongation of shelf life. Overall, compared with soft-flesh pears, the change in 1-Hexanol content in crisp-flesh pears was greater, while the change in Hexanal and (E)-2-Hexanal content was smaller.

Yali, Hanhongli, Ruanerli and Nanguoli had a high concentration of (E)-2-Hexen-1-ol. Generally, its content in Yali and Ruanerli decreased gradually as the shelf life extended, while the highest content was detected in Hanhongli and Nanguoli at the 7 d and 3 d of shelf life, respectively, and then showed a decreasing trend. Monoterpene alcohols such as linalool and α-Terpineol are important aroma and flavor compounds in fruits [43]. Linalool is present in plants mainly as a component of essential oils and is produced by many flowers and spice plants with pleasant floral and citrus aromas [44]. Yali and Ruanerli had a higher content of Linalool, and its content in Yali showed a trend of increasing and then decreasing with the prolongation of shelf life, with the highest content detected at 3 d of shelf life, while Linalool content in Ruanerli fluctuated, reaching its maximum value of 0.08 mg/kg at 0 d of shelf life. It was found that that α-Terpineol is associated with a floral and rose aroma [45]. Ruanerli had higher levels of α-Terpineol compared to other varieties, with the highest level of 0.52 mg/kg detected at 14 d of shelf life. Jingbaili had a high concentration of trans-3-Hexenol on the day of harvest, which provides a green aroma odor to the fruit [46]. Furthermore, both Louis and Cure had higher levels of 1-Octen-3-ol, (E)-2-Octenal, (E)-2-Heptenal, and (E)-2-Decenal throughout the shelf life.

Ketones. Ketones are the major fruity, sweet compounds in fruits [10]. Most ketones are produced via the degradation or decarboxylation pathway of carotenoids [47], with (E)-1-(2,6,6-Trimethyl-1,3-cyclohexadien-1-yl)-2-buten-1-one having a rose-like flavor and honey-like woody aroma [48]. (E)-1-(2,6,6-Trimethyl-1,3-cyclohexadien-1-yl)-2-buten-1-one was detected in all eight pear varieties, with higher levels in Louis, Cure and Ruanerli. 6-Methyl-5-heptene-2-one is reported to be a degradation product of lycopene [42] or α-Farnesene [49] with green color and citrus flavor, which is commonly found in fruits such as apricot [42], lychee [44], prune plum [50] and apple [51]. It was found that Louis and Cure had higher concentrations of 6-Methyl-5-heptene-2-one; its content in Louis showed an increasing and then decreasing trend with the prolongation of shelf life, and the highest content of 0.17 mg/kg was detected at 3 d of shelf life, while its content in Cure gradually decreased.

Hydrocarbons and others. At 14 d and 21 d of shelf life, Jingbaili and Nanguoli had high levels of (E,E)-α-Farnesene, which were significantly higher than the rest of the shelf life. At the same time, Jingbaili also had high content of (Z,E)-α-Farnesene. Other compounds were also detected in the eight pear varieties, such as trans-α,α,5-Trimethyl-5-ethenyltetrahydro-2-furanmethanol, which was found to be higher in Ruanerli, and the content gradually decreased with the prolongation of shelf life. 

The results showed that the content of each VOC did not change much in crisp-flesh pears throughout the shelf life, while significant differences existed in soft-flesh pears. From a comparative analysis of the changes in the content of differential aroma substances in the eight pear varieties during shelf life, it was discovered that Yali and Hanhongli had green and sweet fragrance characteristics, and were best consumed within their respective shelf life of 0–3 d and 14 d. Korla pear exhibited green aroma characteristics and was best consumed within a shelf life of approximately 14 d. Nanguoli and Ruanerli had fruity aroma characteristics, with optimal consumption periods of 14–21 d and 3–7 d, respectively. In comparison, Jingbaili had a lighter fruity aroma, making it more suitable for consumption within a shelf life of 14 d. Louis and Cure exhibited floral and fruity aromas and were best for consumption after 21 d of shelf life.

## 4. Conclusions

A total of 160 VOCs were detected in eight pear varieties, and the VOCs with the highest number and content of total VOC were esters, alcohols and aldehydes, which interacted with each other and collectively affected the feature fragrance of pears. The results of OPLS-DA and HCA were used to classify the eight pear varieties into six groups. Based on VIP > 1 and *p* < 0.05, the content of 31 VOCs was further screened to reach a significant level of difference among samples. In the PCA of pears based on differential VOCs, the cumulative variance contribution of the first two PCs reached 66.7%, and the pear samples were further differentiated using clustering heat map analysis. The content of VOCs and aroma flavor in crispy pears did not change much with the extension of shelf life, while the changes were more obvious in soft pears. Aroma flavor was much stronger and the number and content of volatile substances detected in *P. ussuriensis* cultivars were higher than those of *P. communis* cultivars and crispy pears. With the extension of shelf life, the changes in SSC in different pear varieties differed, and SSC was positively correlated with the content of esters and hydrocarbons in a highly significant way. In conclusion, this study revealed the main volatile components of different pear varieties, clarified the material basis for aroma flavor differences of pear varieties with different flesh textures, and provided a reference for determining the optimal shelf life of different pear varieties concerning aroma evaluation and studying the mechanism of differences in VOCs in the future.

## Figures and Tables

**Figure 1 foods-12-04224-f001:**
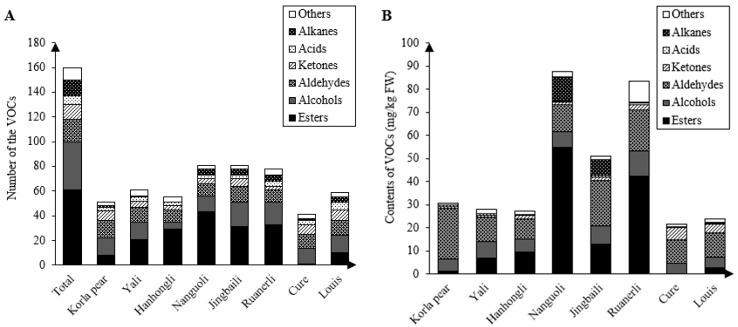
Comparison of the types (**A**) and contents (**B**) of total volatile compounds in eight pear varieties.

**Figure 2 foods-12-04224-f002:**
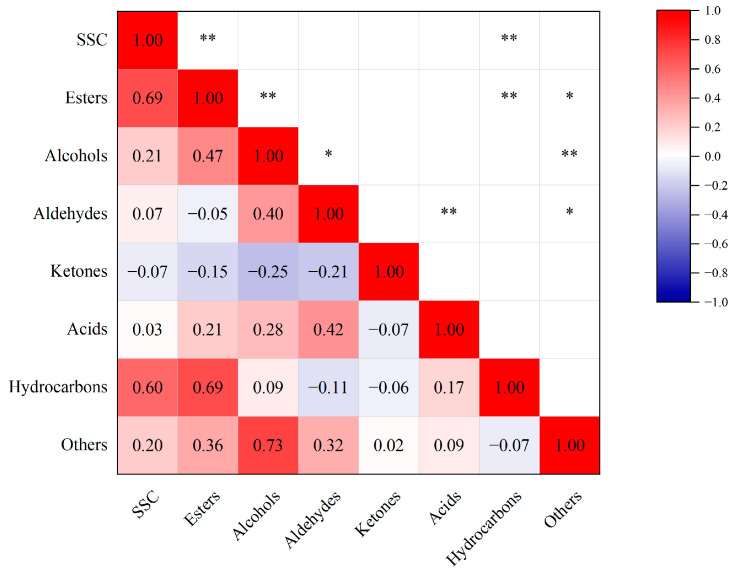
Correlation analysis between SSC and total content of each volatile substance. * indicates significance at *p* < 0.05, ** significance at *p* < 0.01. Color depth indicates Pearson correlation coefficient values.

**Figure 3 foods-12-04224-f003:**
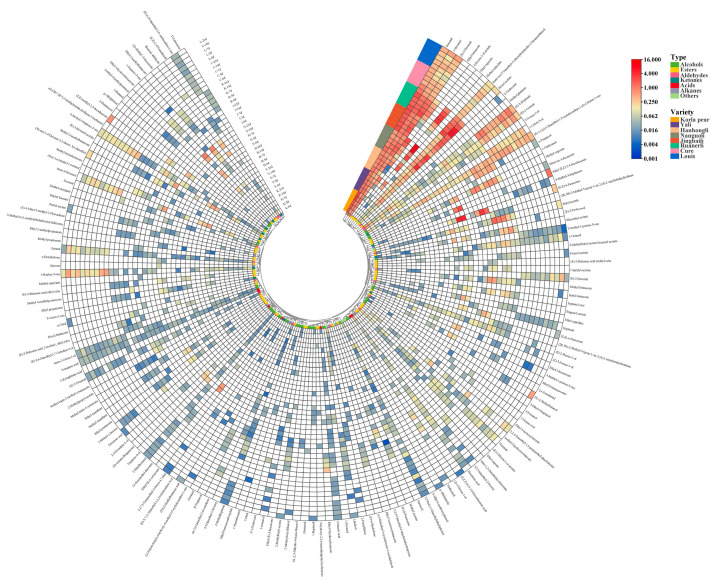
Composition and content of VOCs of the eight pear varieties at different shelf life.

**Figure 4 foods-12-04224-f004:**
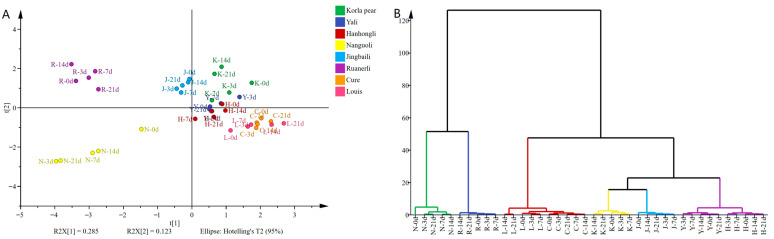
The results of OPLS-DA (**A**), HCA (**B**), model cross-validation results (**C**) and VIP values (**D**) of VOCs of different pear varieties at different shelf life. The bars with red color indicate the differential volatile compounds screened out under the condition with VIP > 1.

**Figure 5 foods-12-04224-f005:**
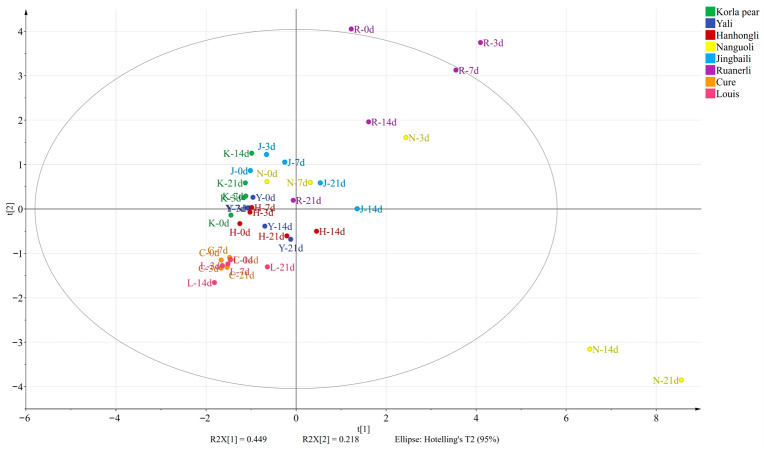
PCA of differential VOCs in pear varieties with different flesh textures.

**Figure 6 foods-12-04224-f006:**
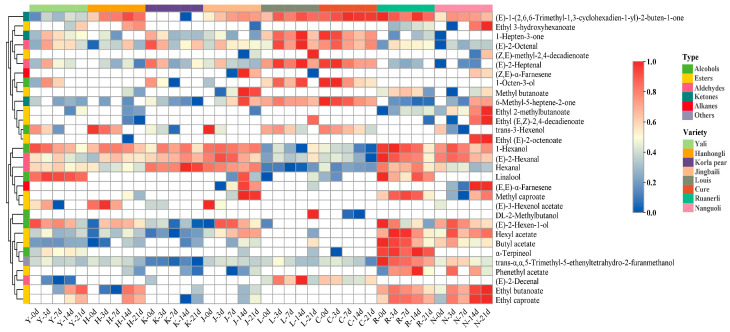
Clustering heat map of differential VOCs in eight pear varieties.

**Table 1 foods-12-04224-t001:** SSC changes of eight pear varieties during shelf life.

Cultivars	Texture	Shelf Life				
		0 d	3 d	7 d	14 d	21 d
Korla pear	Crisp	11.98 ± 0.33 de	12.87 ± 0.44 c	12.99 ± 0.56 cd	11.87 ± 0.72 e	12.70 ± 0.27 d
Yali	Crisp	10.43 ± 0.25 f	10.47 ± 0.34 e	10.21 ± 0.65 f	10.44 ± 0.55 f	10.32 ± 0.36 f
Hanhongli	Crisp	11.61 ± 0.50 e	12.11 ± 0.42 d	12.37 ± 0.95 d	12.59 ± 0.45 d	12.25 ± 0.42 d
Nanguoli	Soft	15.23 ± 0.54 a	16.86 ± 1.26 a	17.04 ± 1.12 a	18.02 ± 1.06 a	17.71 ± 1.04 a
Jingbaili	Soft	12.30 ± 1.46 cd	13.09 ± 0.53 c	13.62 ± 0.83 bc	14.25 ± 0.90 b	14.54 ± 0.86 b
Ruanerli	Soft	14.06 ± 0.57 b	14.28 ± 0.39 b	14.16 ± 0.53 b	13.15 ± 0.75 cd	12.53 ± 1.34 d
Cure	Soft	11.49 ± 0.25 e	11.74 ± 0.57 d	10.94 ± 1.13 e	12.55 ± 0.67 d	11.08 ± 0.39 e
Louis	Soft	12.79 ± 0.41 c	13.13 ± 0.47 c	13.83 ± 0.35 b	13.44 ± 0.70 c	13.82 ± 0.51 c

Note: Values represent mean ± standard deviation (SD) (*n* = 10). Significant differences in results were indicated by different letters (a > b > c > d > e > f).

**Table 2 foods-12-04224-t002:** Types and contents of VOCs in eight pear varieties during shelf life.

Cultivars	Shelf Life	Esters		Alcohols		Aldehydes		Ketones		Acids		Hydrocarbons		Others	
		Types	Contents (mg/kg)	Types	Contents (mg/kg)	Types	Contents (mg/kg)	Types	Contents (mg/kg)	Types	Contents (mg/kg)	Types	Contents (mg/kg)	Types	Contents (mg/kg)
Korla pear	0 d	2	0.17	11	0.98	14	4.75	5	0.26	3	0.35	—	—	3	0.39
	3 d	2	0.23	8	1.33	11	3.81	4	0.15	2	0.09	1	0.03	2	0.13
	7 d	2	0.15	8	0.94	6	3.30	4	0.07	1	0.03	—	—	3	0.10
	14 d	6	0.34	7	1.14	8	5.43	4	0.08	2	0.08	—	—	3	0.14
	21 d	6	0.38	7	0.82	10	4.68	3	0.05	3	0.11	1	0.03	2	0.09
Yali	0 d	5	0.57	8	1.88	8	2.09	4	0.10	2	0.07	—	—	5	0.54
	3 d	4	0.87	10	1.77	11	2.58	3	0.19	3	0.16	1	0.02	4	0.54
	7 d	7	0.39	10	1.32	11	2.66	3	0.16	3	0.10	—	—	3	0.25
	14 d	16	1.54	9	1.24	9	1.80	3	0.15	3	0.11	—	—	2	0.31
	21 d	20	3.48	10	0.92	6	1.61	4	0.16	1	0.03	—	—	1	0.18
Hanhongli	0 d	3	0.41	4	1.20	3	1.40	1	0.13	2	0.03	—	—	2	0.36
	3 d	6	0.98	4	1.44	6	1.65	3	0.35	3	0.10	—	—	2	0.23
	7 d	4	0.51	4	1.30	7	1.85	2	0.30	3	0.10	—	—	3	0.32
	14 d	24	4.72	5	1.06	9	2.15	2	0.44	3	0.15	—	—	2	0.31
	21 d	23	2.92	2	0.72	6	1.56	1	0.31	1	0.02	—	—	3	0.18
Nanguoli	0 d	8	1.12	4	1.15	7	2.76	2	0.11	—	—	—	—	2	0.33
	3 d	22	9.37	10	1.70	7	2.94	2	0.25	3	0.08	3	0.20	2	0.62
	7 d	16	3.44	8	1.25	4	2.41	2	0.23	1	0.03	2	0.10	2	0.61
	14 d	33	17.60	7	1.05	7	1.86	3	0.34	3	0.12	5	5.11	2	0.38
	21 d	39	23.42	6	1.51	5	1.68	2	0.28	3	0.13	6	4.95	2	0.31
Jingbaili	0 d	2	0.92	7	1.77	7	4.09	3	0.15	2	0.10	1	0.02	2	0.23
	3 d	4	0.98	8	1.43	10	5.01	3	0.25	3	0.19	2	0.06	3	0.32
	7 d	11	1.73	8	1.36	10	4.10	4	0.31	3	0.13	3	0.59	3	0.34
	14 d	22	4.63	15	1.63	11	3.22	3	0.49	3	0.13	4	4.38	3	0.31
	21 d	27	4.71	14	1.70	9	3.25	4	0.34	3	0.13	4	1.82	2	0.26
Ruanerli	0 d	16	4.79	12	2.40	6	5.04	1	0.55	2	0.07	3	0.12	5	3.20
	3 d	27	13.76	13	2.87	8	3.73	3	0.41	3	0.18	2	0.06	4	1.82
	7 d	28	12.94	12	2.36	7	3.34	2	0.29	3	0.16	3	0.17	4	1.68
	14 d	28	7.58	10	2.03	6	3.87	3	0.55	2	0.11	3	0.15	3	1.28
	21 d	24	3.27	10	1.30	6	1.88	3	0.38	2	0.05	2	0.13	4	0.98
Cure	0 d	—	—	7	1.16	8	2.74	6	—	1	0.02	1	0.02	2	0.40
	3 d	—	—	9	1.11	10	2.64	7	1.15	3	0.11	1	0.02	2	0.31
	7 d	—	—	7	0.86	9	1.86	6	1.03	3	0.09	—	—	2	0.27
	14 d	1	0.12	9	0.59	9	1.50	6	0.84	2	0.07	1	0.03	2	0.22
	21 d	1	0.06	7	0.49	10	1.63	6	0.88	3	0.07	1	0.06	3	0.11
Louis	0 d	—	—	8	0.66	9	1.22	6	0.43	1	0.03	2	0.02	2	0.35
	3 d	—	—	8	1.06	10	2.69	6	0.92	3	0.09	1	0.02	3	0.39
	7 d	—	—	6	0.71	10	1.70	7	0.68	3	0.10	1	0.02	2	0.28
	14 d	2	0.21	10	0.81	11	3.16	6	1.13	3	0.09	2	0.03	2	0.20
	21 d	9	2.47	9	1.21	9	1.71	6	0.77	5	0.11	2	0.29	3	0.20

Note: — indicates not detected.

## Data Availability

Data will be made available on request from the corresponding author.

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
