# Peer review of "Changes of Volatile Organic Compounds of Different Flesh Texture Pears during Shelf Life Based on Headspace Solid-Phase Microextraction with Gas Chromatography–Mass Spectrometry"

_foods, 2023, doi:10.3390/foods12234224_

Round 1

Reviewer 1 Report

Comments and Suggestions for Authors

 The manuscript " Changes of volatile organic compounds of different flesh texture pears during shelf life based on HS-SPME with GC-MS" is well written, the idea of this paper is very interesting. The title of the manuscript is suitable of researching. Aim of the paper is clear, Introduction is well written.

In Material and Methods (GC-MS analysis) authors should provide information on the number of repetitions of individual analyses and information about retention time (RT) and retention indices (RI) of volatile compounds.

The conclusion is connected with the obtained results and the contribution of the research is highlighted.

Check spelling and grammar throughout the manuscript.

Author Response

Thank you very much for your valuable comments on improving the quality of this manuscript. The manuscript has been carefully revised and proofread according to the reviewer's comments. The response to the reviewer's comments is the following:

Comments 1: In Material and Methods (GC-MS analysis) authors should provide information on the number of repetitions of individual analyses.

Response 1: Thanks for your professional suggestion. Information on the number of individual analytical replicates has been provided in lines 129 through 130 of Materials and Methods (GC-MS analysis).

Comments 2: In Material and Methods (GC-MS analysis) authors should provide information about retention time (RT) and retention indices (RI) of volatile compounds.

Response 2: Thanks for your comments. Information on retention time (RT) of volatile organic compounds has been provided in lines 126 through 129 of Materials and Methods (GC-MS analysis). The retention indices (RI) qualitative analysis method uses a series of RI benchmark substances (e.g., n-alkanes) as a reference for identification of volatile compounds. However, in our study, the standard mass spectra from the NIST 17s database were used to retrieve and manually analyze the spectral data obtained from the experiments, and a match of more than 90% was selected as the basis for the identification of volatile compounds. We also note that Chenchen Wang et al. (Ref. 6) and Guanwei Gao et al. (Ref. 10) also used this method for the identification of volatile organic compounds, and therefore we do not provide information on the RI of volatile compounds in our Materials and Methods (GC-MS analysis).

Comments 3: Check spelling and grammar throughout the manuscript.

Response 3: Thanks for your careful checks. We agree with this suggestion and have made the corrections to make the word harmonized within the whole manuscript.

Based on the reviewer's suggestions, we have made the following changes:

  1. In line 29, correct "hm2" to "hm2".
  2. In line 35, correct "Pyrus communis" to "P. communis" and set it in italics.
  3. In line 69, correct " and changes in aroma substances were important in determining the shelf life of fruits." to " and changes of aroma substances were important in determining the shelf life of fruits.".
  4. In line 132, correct " volatile organic compound" to " VOCs ".
  5. In line 197, correct " Correlation analys was between SSC and total content of each volatile substance." to " Correlation analysis was between SSC and total content of each volatile substance. ".

Thank you again for your positive comments and valuable suggestions to improve the quality of our manuscript.

Yours sincerely,

Luming Tian

Reviewer 2 Report

Comments and Suggestions for Authors

line 28 through 76: The paper turns out to be clear and concise, the objective of the work respected and completed.

The introduction makes a clear excursus on the botany of pears, varieties and how they, are distinguished by their aromatic components. Materials and methods 78 to 142 results in a fluent description of the different analyses that are performed for the different parameters under consideration.       

Results and discussion i.e. 144 to 397; Thorough and well done explanations of the presence of the recognized VOCs by variety. In line 374 it would be interesting to compare the VOCs found, particularly the Ketones and how they determine citrus flavor to some sensory analysis performed on the same varieties: Loouis, Cure and Runarerli.

I believe parallels could be made between VOCs and sensory analyses done on pears

 Conclusions i.e., 399 to 416 meet the objective set from the beginning, which is the evaluation of aroma components in different pear varieties.

Author Response

We appreciate you very much for your positive and constructive comments on our manuscript. The response to the reviewer's comments is the following:

Comments: Line 28 through 76: The paper turns out to be clear and concise, the objective of the work respected and completed. The introduction makes a clear excursus on the botany of pears, varieties and how they, are distinguished by their aromatic components. Materials and methods 78 to 142 results in a fluent description of the different analyses that are performed for the different parameters under consideration. Results and discussion i.e. 144 to 397; Thorough and well done explanations of the presence of the recognized VOCs by variety. In line 374 it would be interesting to compare the VOCs found, particularly the Ketones and how they determine citrus flavor to some sensory analysis performed on the same varieties: Louis, Cure and Ruanerli. I believe parallels could be made between VOCs and sensory analyses done on pears. Conclusions i.e., 399 to 416 meet the objective set from the beginning, which is the evaluation of aroma components in different pear varieties.

Response: We would like to thank you for carefully reading our paper and providing positive comments. Your recognition of our work has given us great confidence and inspired us to engage in further research in this field.

Once again, thank you very much for your comments and suggestions.

Yours sincerely,

Luming Tian

Reviewer 3 Report

Comments and Suggestions for Authors

The authors present a study made on different varieties of pears, three crispy pears, and five soft pears during their shelf life. They used headspace solid-phase microextraction (HS-SPME) combined with gas chromatography-mass spectrometry (GC-MS) to analyze the content and number of volatile organic compounds (VOCs), and the changes of soluble solids content (SSC) were analyzed.

The study is complex and complete by analyzing changes in the SSC of eight pear varieties during shelf life, differences in the composition of total volatile compounds, the correlation analysis between SSC and total content of each volatile compound, changes of VOCs in the flesh of eight pear varieties during different shelf life, and analysis of key VOCs in eight pear varieties based on OPLS-DA and HCA. They also present a differentiation analysis of the pulp texture of pear cultivars based on differentiated VOCs, and an analysis of differential VOCs in the flesh of eight pear varieties.

The results are well presented and explained, and the conclusions support them.

Author Response

We feel great thanks for your professional review work on our article. The response to the reviewer's comments is the following:

Comments: The authors present a study made on different varieties of pears, three crispy pears, and five soft pears during their shelf life. They used headspace solid-phase microextraction (HS-SPME) combined with gas chromatography-mass spectrometry (GC-MS) to analyze the content and number of volatile organic compounds (VOCs), and the changes of soluble solids content (SSC) were analyzed. The study is complex and complete by analyzing changes in the SSC of eight pear varieties during shelf life, differences in the composition of total volatile compounds, the correlation analysis between SSC and total content of each volatile compound, changes of VOCs in the flesh of eight pear varieties during different shelf life, and analysis of key VOCs in eight pear varieties based on OPLS-DA and HCA. They also present a differentiation analysis of the pulp texture of pear cultivars based on differentiated VOCs, and an analysis of differential VOCs in the flesh of eight pear varieties. The results are well presented and explained, and the conclusions support them.

Response: We thank the reviewers for your professional review work and positive feedback on our manuscript and for accurately summarizing the results of the paper. We are encouraged by your recognition of our work.

Once again, thank you very much for your comments and suggestions.

Yours sincerely,

Luming Tian